# Nano-Agrochemicals as Substitutes for Pesticides: Prospects and Risks

**DOI:** 10.3390/plants13010109

**Published:** 2023-12-29

**Authors:** Shehbaz Ali, Naveed Ahmad, Mudasir A. Dar, Sehrish Manan, Abida Rani, Suliman Mohammed Suliman Alghanem, Khalid Ali Khan, Sivasamy Sethupathy, Noureddine Elboughdiri, Yasser S. Mostafa, Saad A. Alamri, Mohamed Hashem, Muhammad Shahid, Daochen Zhu

**Affiliations:** 1Biofuels Institute, School of Emergency Management, School of the Environment and Safety Engineering, Jiangsu University, Zhenjiang 212013, China; shehbaz205@gmail.com (S.A.); muddar7@ujs.edu.cn (M.A.D.); sehrish_manan@hust.edu.cn (S.M.); sethulifescience@gmail.com (S.S.); 2Joint Center for Single Cell Biology, Shanghai Collaborative Innovation Center of Agri-Seeds, School of Agriculture and Biology, Shanghai Jiao Tong University, 800 Dongchuan Road, Shanghai 200240, China; naveedjlau@gmail.com; 3Department of Pharmaceutical Chemistry, Faculty of Pharmacy, Bahauddin Zakariya University, Multan 60800, Pakistan; drabida.rani@gmail.com; 4Department of Biology, College of Science, Qassim University, Burydah 52571, Saudi Arabia; su.alghanem@qu.edu.sa; 5Applied College, Mahala Campus and the Unit of Bee Research and Honey Production/Research Center for Advanced Materials Science (RCAMS), King Khalid University, P.O. Box 9004, Abha 61413, Saudi Arabia; kkhan@kku.edu.ksa; 6Chemical Engineering Department, College of Engineering, University of Ha’il, P.O. Box 2440, Ha’il 81441, Saudi Arabia; ghilaninouri@yahoo.fr; 7Chemical Engineering Process Department, National School of Engineers Gabes, University of Gabes, Gabes 6029, Tunisia; 8Department of Biology, College of Science, King Khalid University, P.O. Box 9004, Abha 61413, Saudi Arabia; ysmosutafa@kku.edu.sa (Y.S.M.); saralomari@kku.edu.sa (S.A.A.); 9Department of Botany and Microbiology, Faculty of Science, Assiut University, Assiut 71515, Egypt; mhashem@aun.edu.eg; 10Department of Bioinformatics and Biotechnology, Government College University, Faisalabad 38000, Pakistan

**Keywords:** agriculture, nanomaterials, herbicides, pesticides, plant disease

## Abstract

This review delves into the mesmerizing technology of nano-agrochemicals, specifically pesticides and herbicides, and their potential to aid in the achievement of UN SDG 17, which aims to reduce hunger and poverty globally. The global market for conventional pesticides and herbicides is expected to reach USD 82.9 billion by 2027, growing 2.7% annually, with North America, Europe, and the Asia–Pacific region being the biggest markets. However, the extensive use of chemical pesticides has proven adverse effects on human health as well as the ecosystem. Therefore, the efficacy, mechanisms, and environmental impacts of conventional pesticides require sustainable alternatives for effective pest management. Undoubtedly, nano-agrochemicals have the potential to completely transform agriculture by increasing crop yields with reduced environmental contamination. The present review discusses the effectiveness and environmental impact of nanopesticides as promising strategies for sustainable agriculture. It provides a concise overview of green nano-agrochemical synthesis and agricultural applications, and the efficacy of nano-agrochemicals against pests including insects and weeds. Nano-agrochemical pesticides are investigated due to their unique size and exceptional performance advantages over conventional ones. Here, we have focused on the environmental risks and current state of nano-agrochemicals, emphasizing the need for further investigations. The review also draws the attention of agriculturists and stakeholders to the current trends of nanomaterial use in agriculture especially for reducing plant diseases and pests. A discussion of the pros and cons of nano-agrochemicals is paramount for their application in sustainable agriculture.

## 1. Introduction

The projected increase in the global population from 7.35 billion to 8.60 billion by the year 2030 poses significant challenges for food security planning. Crop production needs to be increased by up to 70% in order to fully meet the current crop demand [1]. The United Nations has set a goal of ending global hunger and poverty as part of its 2030 sustainable development goals (SDGs) 17 agenda. A more productive, environmentally friendly, and expansive agricultural sector is necessary to meet development goals in the face of climate change and finite resources [2]. The agricultural sector is currently confronting a wide range of challenges, including decreased crop productivity, soil nutrient deficiencies, the impact of climate change, limited water resources, declining soil fertility, organic matter decomposition in soil, crop ailments, a lack of understanding of genetically modified organisms, and insufficient workforce [3].

The enhancement of crop productivity can be achieved through the implementation of transgenic crop development and the utilization of marker-assisted breeding techniques. These advancements also exert adverse effects on the environment, soil fertility, and the ability of plants to resist viruses and pests [1,4]. On the other hand, the optimal crop yield is contingent upon the utilization of improved varieties, effective pest and disease control measures, and recommended fertilization practices. Effective pest management is a crucial determinant for cultivating robust and productive crops that can adequately sustain the growing population’s nutritional needs [5]. It is estimated that over 50% of the chemical fertilizers and pesticides utilized are lost as a result of leaching or mineralization [1,4]. In the last seven decades, traditional agriculture has been plagued by an overreliance on synthetic chemicals like pesticides to control pests, aiming to meet the food requirements of a rapidly growing population [6]. The persistence of these chemicals and the generation of toxic intermediates during their degradation have adverse impacts on both the biotic and abiotic components of the biosphere [7]. These chemical components have polluted the air, soil, water, and vegetation, bioaccumulated in the food chain, and posed a threat to the lives of non-target animals and plants; for example, chlorpyrifos is found in urban streams and is the main cause of death of aquatic organisms and fishes or invertebrates [8,9]. The use of synthetic chemicals also causes a steady decline in insect populations, damages communities of soil microorganisms, and results in an overall decline in biodiversity [9]. Additionally, negative effects on human health, including cancer, reproductive problems, and neurological disorders have been reported [10]. Weeds are a major source of biotic stress and loss of crop yield in agriculture, reducing harvests by an estimated 20–90%. Farmers often use conventional herbicides, which are highly toxic not only to plants and animals, but also to humans through breathing contaminated air, skin contact, or ingestion [11]. Consequences of conventional herbicides include reduced growth (reduced photosynthetic activity and amino acid synthesis) and reproduction (cellular division synthesis), as well as elevated mortality rates among various plant species such as macrophytes, periphyton, and phytoplankton [11,12]. All these serious environmental and health issues require a significant effort to be made to create ecofriendly and healthy solutions with the aim of augmenting food production within the constraints of limited resources. Hence, the present moment presents an opportune occasion to devise innovative approaches, such as nanotechnology, to facilitate the sustainable advancement of agriculture [1]. 

Nano-agrochemicals (NACs) are nanomaterials and formulations specifically designed and controlled at the nanoscale [13,14]. They can be defined as agricultural formulations in the form of nanopesticides (NPCs) which have unique properties (high aspect ratio) due to elements ranging in nanometer size (up to 100 nm) [2,3,15]. They have the potential to revolutionize agriculture through the promotion of efficient and ecofriendly NPCs with properties such as target specificity, the ability to control usage, and high surface area which allow the targeted delivery of nutrients and protection of crop yields [13,16]. The latest scientific investigations have unveiled a remarkable increase in efficiency, measuring at an impressive 31.5% higher than the established standards of conventional pesticides [17]. Therefore, NPCs overcome the limitations of conventional pesticides, i.e., limited bioavailability, vulnerability to light-induced degradation, and the harmful consequences of organic solvent pollution [13]. The benefits and prevailing factors of utilizing NPCs are succinctly outlined in Figure 1, which serves as a visual depiction of the factors associated with NPCs and their inherent advantages [18]. 

The regulations surrounding the use of pesticides are intricate. Safety standards for consumers, the environment, and people’s health are all enforced by governments and regulatory agencies. Pesticide-related businesses have to deal with issues like regulatory compliance, residues, and environmental impact [19,20]. It is estimated that pests and weeds cause a global average loss in crop yields of 35% [21]. The global market for pesticides has expanded as a result of the attempt to control pests and weeds. The modern farming market in the Asia–Pacific and Latin American regions has already reached a value of billions in USD [22]. It was estimated in 2009 that agriculture producers globally supply approximately USD 40 billion worth of pesticides annually, which are used to eliminate pest-related diseases worldwide [23]. The global use of pesticides significantly increased from 1990 to 2007, and this trend then changed after 2007 and 2014 [24]. Although there are many classes of pesticides, some internationally used ones are (1) herbicides (phenoxy hormone products, triazines, amides, carbamates herbicides, and dinitroanilines), (2) insecticides (chlorinated hydrocarbons, organo-phosphates, carbamates, pyrethroids, carbamates-insect-SdTr, and pyrethroids-SeedTr-Ins), and (3) fungicides and bactericides (dithiocarbamates, benzimidazoles, triazoles, and dialzenes) [25]. The utilization of pesticides across continents, the top 10 countries with the highest consumption of pesticides per unit area of cropland (kg/ha), and the percentage of usage for agricultural purposes from 2000 to 2021 are depicted in Figure 2 [26]. 

The pesticide market, encompassing herbicides, insecticides, and fungicides, yields substantial annual profits on a global scale. The projected growth of the global pesticide market is estimated to be from USD 78.16 billion in 2021 to USD 85.11 billion in 2022, with a compound annual growth rate (CAGR) of 8.90%. The market is expected to reach a value of USD 105.39 billion by 2026, with a CAGR of 5.5% [27]. The global market for herbicides reached a significant valuation of USD 31.50 billion in 2022 [28]. The projected growth of the market is expected to increase from USD 33.39 billion in 2023 to USD 53.21 billion by 2032 [29]. The global market share value of insecticides in 2022 was USD 19.50 billion [30]. The projected growth of the global insecticide market is estimated to be from USD 19.50 billion in 2022 to USD 20.95 billion in 2023, with a CAGR of 7.4% [27]. The global market for fungicides reached a significant valuation of USD 16.35 billion in 2019 [31]. The projected growth of the fungicide market is expected to reach USD 25.81 billion by 2028, with a CAGR of 4.3% from its previous value of USD 18.43 billion in 2021 [32]. According to an alternative estimation, the projected value of the worldwide fungicide market is anticipated to reach USD 41.9 billion by the year 2032, exhibiting a CAGR of 7.0% [33]. The primary factors driving the growth of this industry are a growing need for food to feed the world’s rapidly expanding population, constraints on the availability of arable land, and a heightened awareness of the importance of pesticide use. Additionally, pesticide companies use a variety of strategies to meet rising demand, increase competitive advantage, foster product and technological innovation, reduce production costs, and expand their customer base. The decision of farmers to abstain from the use of pesticides carries the risk of diminishing crop yield, which in turn could have adverse effects on both the food chain and the availability of agricultural goods [34]. 

### 1.1. Effectiveness of Pesticides

The efficacy of pesticides in pest control is contingent upon various factors, encompassing the particular pest species, the type of pesticide employed, the severity of the infestation, the method of application, and prevailing environmental conditions. Pesticides are specifically formulated to show toxicity towards targeted pests, encompassing a wide range of organisms such as insects, plant diseases, weeds, and other undesirable entities [9,35,36]. The choice of a pesticide ought to be contingent upon the particular circumstances at hand, with due regard for the potential ramifications on non-target organisms during the selection process. In the context of chronic pest problems, it is advisable to employ spot treatments and opt for less toxic substances that have a longer duration of efficacy. Conversely, in emergency scenarios, it may be imperative to utilize substances that have a shorter lifespan, act rapidly, and possess higher acute toxicity [35]. In controlled laboratory settings, pesticides often exhibit significant efficacy (Table 1). However, their effectiveness can be diminished in real-world scenarios due to various factors, including the presence of natural predators, fluctuations in temperature, and other environmental conditions [9,35,36]. Factors affecting the effectiveness of pesticides and their classification based on their nature and applications are summarized in Figure 3. 

### 1.2. Ecological Risks of Pesticides 

Exposure to pesticides can result in a range of negative outcomes, varying from minor skin irritation or allergic reactions to more pronounced and severe symptoms. Certain insecticides, although posing limited harm to human health, can display substantial toxicity towards beneficial insects, such as parasitic wasps and honeybees, as well as other desirable organisms like earthworms and aquatic invertebrates [9,35]. When comparing herbicides, fungicides, and insecticides, it is evident that insecticides display a greater level of toxicity towards the environment. It is important to highlight those specific herbicides that demonstrate considerably greater levels of hazards and toxicity compared to insecticides, which are primarily attributed to variances in solubility. Water-soluble compounds have the ability to be transported into groundwater, rivers, streams, and lakes, subsequently entering living organisms and persisting within the food chain [85]. The persistence of pesticide residues in groundwater is prolonged. Alachlor, atrazine, and aldicarb are frequently encountered pesticides in groundwater [86]. The careful choice of pesticides with minimal toxicity to mammals, quick biodegradation, and limited harm to non-target organisms is crucial in order to mitigate any potential adverse effects [87]. 

Ecologically toxic pesticides which may pollute the environment are herbicides (paraquat, glyphosate, atrazine, and 2, 4-D), insecticides (parathion, diazinon, aldicarb, and DDT) and fungicides (captan, benomyl, and copper) [88]. Herbicides can kill some beneficial plant species which provide food and shelter for wildlife species [89]. An insecticide’s potential to kill the targeted species may also affect invertebrates which are a food source for birds [86]. Due to the toxicity and extensive application of pesticides, amphibians are among the most threatened species on earth [89]; pesticides also threaten human health and aquatic environments [54,55]. With significant concerns regarding toxicity, the WHO has defined pesticide residue as any substance in the food of living organisms originating from pesticides and regarded as toxic compounds [9,10]. Long-term pesticide exposure may lead to a few types of cancer like prostate and lung, and neurological disorders such as Parkinson’s disease and Alzheimer’s [9,10]. Consumers worldwide are progressively demanding food free of hazardous pesticide residues [9,10]. Different countries have started programs to analyze pesticide remnants in food and several countries are applying programs to reduce pesticide use to minimize their hazardous impact [90]. For example, atrazine, owing to its ease of use, affordability, effectiveness in controlling weeds, and relatively extended persistence in the environment, is frequently detected as one of the predominant herbicides in water sources. Although it is banned in many countries such as those of the European Union (since 2004) and China (since 2012), and is partially banned in Australia and Canada, it is extensively utilized in the United States and frequently detected in water sources exceeding the limit set by the Environmental Protection Agency (EPA). Exposure to atrazine in both humans and animals has been associated with cardiovascular alterations, the specific underlying cause of which remains undefined [91,92]. The effectiveness of different pesticides and their ecological risks are listed in Table 1. 

### 1.3. Recommendations for the Application of Pesticides 

There are several prospective avenues for future development which hold great promise for the creation of ecologically sustainable and efficient methods of delivering pesticides. These include the following approaches.

Precision agriculture, also known as smart farming or satellite farming, is an innovative approach that utilizes advanced technologies to optimize agricultural practices. By integrating various tools such as a combination of sensors, data analytics, and GPS technology, this system collects comprehensive data pertaining to agricultural fields. The data collected encompass crucial aspects such as soil conditions, crop health, and levels of pest infestation. The data are utilized to tailor agricultural methodologies, such as the application of pesticides, to suit the specific characteristics of individual fields. This approach provides target-specific application, mitigates environmental pollution and safeguards non-target organisms, and optimizes timing and data-driven decision making for pesticide application based on real-time data [93]. 

Smart delivery systems encompass pesticide application technologies that employ precise release mechanisms to administer pesticides at specific times and rates. These systems maximize the efficacy of pesticides while minimizing their impact on the environment. Microcapsules represent an intelligent and efficient method of delivering pesticides. Microcapsules, minute spherical structures, envelop pesticides within a protective layer. The gradual degradation of the coating facilitates the controlled release of the pesticide. This mechanism guarantees the controlled release of the pesticide, optimizing its efficacy while minimizing the potential for runoff or volatilization hazards. Intelligent agents exemplify a sophisticated delivery system. Intelligent agents, in the form of micro-robots, strategically deliver pesticides to targeted areas within plant tissues or soil. This technology facilitates the targeted application of pesticides, reducing the potential for unintended harm to surrounding organisms [93].

Biomimetic pesticide delivery systems are innovative approaches inspired by nature to efficiently transport pesticides. This approach seeks to enhance the efficacy, efficiency, and ecofriendliness of pesticide delivery systems. Wax-based coatings exemplify a biomimetic approach for delivering pesticides. Pheromones possess the potential to serve as a biomimetic mechanism for delivering pesticides. Pheromones are bioactive compounds secreted by organisms to facilitate inter-species communication. They lure pests towards traps or designated areas for the purpose of pesticide application. Encapsulation is a process in which pesticides are enveloped in a protective coating. Coatings can be fabricated using a variety of materials, including polymers, waxes, and hydrogels. Encapsulation serves as a protective barrier for the pesticide, enhancing its efficacy while minimizing the potential for environmental contamination [94]. The utilization of cutting-edge technologies holds great potential to enhance pest management and safeguard crop health. This practice fosters sustainable agricultural methods and contributes to the enhancement of environmental well-being.

## 2. Alternatives

The drawbacks of conventional pesticides which harm the environment and people’s health, as well as contribute to pest resistance, dictate the use of bio-based pesticides, NACs, and other alternatives. Researchers are now looking into alternatives that are safer and more effective at eradicating pests.

### 2.1. Biopesticides

Biopesticides (BPs), derived from various organic sources such as plants, animals, insects, fungi, and microorganisms, are considered to be safer and more environmentally friendly alternatives in pest control. Currently, BPs hold a 5% market share in the global pesticide industry and are used specifically for the purpose of environmentally sustainable pest management [95]. BPs are widely regarded as a potential remedy for addressing the challenges associated with conventional pesticides. The liquid-based formulation of these substances is characterized by its simplicity, making it convenient to handle. These products possess environmentally friendly characteristics, as they are biodegradable and non-toxic, and exhibit target specificity, rendering them highly effective even in small quantities [96]. The first documented evidence of the use of BPs related to the identification of mycoherbicides (fungi-based herbicides) in the middle of the 1970s. When bioherbicides were first introduced to the market in 1980, they were only used by farmers in the US, Canada, Ukraine, and Europe [97]. BPs can be classified into three primary categories.

Microbial biopesticides (MBPs) are composed of naturally present microorganisms (bacteria, fungi, viruses, and protozoans). MBPs control pests by following natural mechanisms and share many of the characteristics of pesticides. They cope with problems like insect resistance and target modification. More than 3000 microbes which infect and kill insects are available for integrated pest management. *Bacillus thuringiensis* (Bt) currently dominates approximately 90% of the MBP market. However, other organisms such as *Beauveria bassiana*, *Baculovirus*, *Steinernema*, *Nosema*, and *Chlorella* have also shown noteworthy contributions in this field [95]. Currently, the identification of natural enemies of insect pests remains limited, with only 15% of these organisms having been successfully recognized and documented. The primary contributors for MBPs are parasitoids belonging to the Hymenoptera order, as well as predators from the Neuroptera, Hemiptera, and Coleoptera orders. Globally, there exists a wide array of over 125 species of natural enemies that are commercially accessible for the purpose of implementing biological control programmes. Some notable examples include *Trichogramma* spp., *Encarsia formosa* Gahan, and *Phytoseiulus persimilis* Athias-Henriot [5]. There are also some pesticides which use fungi (*Aschersonia aleyrodis*) and protozoa (*Nosema locustae*) as active ingredients [96]. 

Natural and synthetic substances with active ingredients that effectively manage pests through mechanisms that are not toxic to the targeted pests, the surrounding environment, or humans are collectively referred to as biochemical biopesticides (BCBPs) [95]. These substances range from essential oils and semiochemicals to plant growth regulators and insect growth regulators to secondary metabolites and natural minerals, all of which are frequently used as BCBPs [98]. These BCBPs typically exhibit a preference for particular pest species. Their composition potentially includes insect pheromones which have the ability to interfere with the mating behaviors of pests. Additionally, it may consist of botanical extracts possessing insecticidal or herbicidal characteristics, such as neem oil or pyrethrins sourced from chrysanthemum flowers [95,98,99]. The term “Plant Incorporated Protectants” (PIPs) is used to describe a wide range of plant-derived substances, both naturally occurring and genetically modified, that have the ability to control or eradicate pests. Genes for resistance to pests and diseases, such as Bt genes, chitinase, lactinase, and protease inhibitors, can be introduced into plant genomes through molecular techniques. The modified genes are expressed in these plants to produce chemicals with pesticide effects [9,10,95]. Indigenous PIPs are derived from botanicals like garlic (*Allium sativum*), cassia (*Cinnamomum cassia*), neem (*Azadirachta indica*), pine (Pinus), and triphala (a blend of three fruits). Proteins found in plant-based products are subject to biodegradation, a process that breaks them down into simpler building blocks. The fact that these proteins are safe for consumption by humans and animals is a major plus [10,95].

### 2.2. Effectiveness of Biopesticides

BPs may be less effective against a broader range of agricultural issues because they are more focused on specific pests or weeds than conventional pesticides. They are also influenced by environmental factors; these interventions necessitate precise timing and optimal application conditions [2,100,101]. They may also necessitate more frequent applications to keep problems under control, increasing labor costs and energy consumption in terms of fuel [101]. They have short shelf lives and may need to be refrigerated or frozen to remain effective. Inadequate research has been conducted on the long-term efficacy, safety, and consequences of certain biopesticides, which is a major concern. The safety and longevity of BPs are dependent on ongoing research and monitoring [2,100,101]. Their effectiveness can be influenced by the formulation and application techniques employed, as well as the particular pests being targeted. This is the reason why current R&D efforts are primarily directed towards augmenting the durability and utilization of these methods within the framework of integrated pest management (IPM) strategies [2,100]. The effectiveness against target pests and ecological risks of different biopesticides in agriculture are listed in Table 2.

### 2.3. Ecological Risks of Biopesticides 

BPs are derived from naturally occurring sources and are widely acknowledged as being more sustainable options in comparison to conventional synthetic pesticides. However, they also pose moderate ecological risks, as they may result in adverse effects on non-target organisms such as beneficial insects, avian species, aquatic life, and other forms of wildlife [102]. They can enter the environment through a number of different pathways, including spray drift, runoff, and leaching into groundwater. Severity is dependent on type, dosage, quantity, and environmental factors [102]. The biopesticide *Bacillus thuringiensis*, derived from a soil-dwelling bacterium, can persist in the environment and harm non-target insects [96]. Additionally, there is a chance that pests will develop resistance to BPs, making it more difficult to control pests in the future. One such example is glufosinate resistance which has already developed [103]. Despite these risks, they are more environmentally friendly than synthetic pesticides. They can be used in IPM programs to reduce pesticide use while maintaining crop yields because they work well in small quantities, decompose quickly, and are effective. To reduce the risk to the environment from spraying, best practices can be used to minimize spray drift, runoff, and leaching [96,102,104].

**Table 2 plants-13-00109-t002:** Composition, effectiveness, and ecological risk of different biopesticides.

Biopesticide (Source)	Target Pest	Effectiveness	Ecological Risk	References
**Bacterium-based**				
Cry toxins (*Bacillus thuringiensis*)	Caterpillars, beetles, flies	Extremely useful for controlling mosquitoes, caterpillars, certain types of beetles, flies, and black flies, among other pests	Only affects the targeted pests and has no effect on other animals	[105,106]
Serenade Rhapsody (*Bacillus subtilis*) MBI 600, and D747 (*Bacillus amyloliquefaciens*)	Fungal and bacterial phytopathogens, aphicidal, biofertilizer	Extremely potent against a wide range of bacterial and fungal plant pathogens	Environmentally benign and harmless to non-target organisms	[105,107,108,109]
Phenazine-1-carboxylic acid (PCA), phenazine *Pseudomonas fluorescens*	Insecticidal, acaricidal, antimicrobial	Highly effective against phytopathogenic fungi and nematodes	Environmentally benign and harmless to non-target organisms	[105,107,110]
Nonactin, Antimycin A3a, Antimycin A8a, and Antimycin A1a (*Streptomyces spp*) Paenimyxin (*Paenibacillus* spp.)	Fungal plant pathogens, nematodes	Highly effective against phytopathogenic fungi and nematodes	Generally safe for non-target organisms and the environment	[1,3]
Spinosad (*Saccharopolyspora spinosa*)	Caterpillars, thrips, leafminers, fruit flies, borers, beetles	Broad-spectrum insecticide, targeting caterpillars, thrips, leafminers, fruit flies, borers, and beetles	Generally safe for mammals and beneficial insects, but bees and certain beneficial insects can be highly affected by direct exposure to it	[111,112]
**Fungus-based**				
Beauvericin (*Beauveria bassiana, Metarhizium anisopliae* var. *anisopliae, Metarhizium anisopliae var. acridum, Metarhizium anisopliae*)	Aphids, thrips, beetles, spider mites, grubs	Efficiently controls diverse pests	Targets pests without harming beneficial organisms like insects, birds, and mammals	[113,114]
*Aschersonia aleyrodis, Lecanicillium lecanii, Isaria fumosorosea*	Aphids, whiteflies, and thrips	Targets whiteflies, thrips, aphids, and select beetles	Ecologically safe, non-harmful to non-target insects	[96]
**Plant-based**				
Pyrethrum (*Chrysanthemum cinerariifolium*)	Various insects	Broad-spectrum insecticide with rapid action of neurotoxin	Safe for mammals but harmful to beneficial insects and aquatic life	[115]
Azadirachtin (*Azadirachta indica*)	Insecticides	Effectively controls insect	Moderate to high toxicity to aquatic organisms	[116]
Rotenone (*Lonchocarpus* spp.)	Slugs and snails	Broad-spectrum insecticide	Toxic to amphibians and macroinvertebrates; mammals may be at risk	[117]
Ryania (*Ryania speciosa*)	Fruit borers, codling moths, Bollworm	Good to moderate control against these target pests	Toxic to mammals and fish, and can also harm beneficial insects	[118]
Nicotine (*Nicotiana tabacum*)	Aphids, leafhoppers, whiteflies	Effective against aphids, leafhoppers, and spider mites	Harms beneficial insects and mammals	[119]
Capsaicin (*Capsicum* spp.)	*Tribolium castaneum*	Potent insecticide against *Tribolium castaneum*	Safe for beneficial creatures	[120]
Garlic oil (*Allium sativum*)	Various pests	Efficiently repels various pests	Generally safe for beneficial insects, birds, and mammals	[121]
Citronella (*Cymbopogon nardus* and *Cymbopogon winterianus*)	Various pests, particularly mosquitoes			
*Cinnamaldehyde* (*Cinnamomum* spp.)	Aphids, spider mites, thrips			
Eugenol (*Syzygium aromaticum*)				
Thymol (*Thymus vulgaris*)				
Geraniol (*Geraniums and lemongrass*)				
Limonene (*Citrus fruit* spp.)				

## 3. Nano-Agrochemicals (NACs)

NACs, or nanotechnology-based agrochemicals, have gained considerable attention in recent years due to extensive research and development efforts. These cutting-edge products harness the special qualities of NPs to improve the effectiveness, safety, and ecofriendliness of conventional agrochemicals. This technology has emerged as a valuable tool in the agricultural sector, offering novel and efficient solutions for conventional agricultural methods and practices. Some of these include NPCs, among other NACs, which have the capacity to revolutionize agriculture by enhancing sustainability and efficiency [3,13,17,122]. These NPCs, ranging in size from 1 to 200 nm, serve as a vehicle to transport agrochemical ingredients (AcI) [123]. With unique properties, they outperform conventional pesticides by 31.5% [17] in managing crop pathogens, weeds, and insects [123]. These unique properties include enhanced water solubility, improved bioavailability, and increased protection of agrochemicals from environmental degradation [123]. Thus, NPCs address the drawbacks of conventional pesticides such as limited availability, susceptibility to degradation from light, and the negative effects of organic solvent pollution [13].

The research in this field focuses on enhancing green synthesis and ecofriendly methods to minimize the use of expensive and hazardous materials in the synthesis of NACs. Using plants, enzymes, and microorganisms for biogenic synthesis allows better control of the size and shape of nanomaterials. Moreover, recent developments and trends in the field of biology encompass a variety of advancements and emerging patterns in the green synthesis of NACs [3,85,124,125]. During the synthesis process, plant extracts (reducing agents) donate electrons to metal ions in metal salts and reduce them to form metals for NACs under controlled conditions such as temperature, pH, and reactant concentration [124,126]. NPs can be further modified by adding functional groups or coatings to improve their stability, solubility, or targeting ability [127]. Recent studies on biomimetic methods suggest that NPs with built-in pesticide properties can be made from silica-based materials that resemble plant cell walls [128]. Figure 4a shows the schematic green synthesis of NACs. They have the potential to revolutionize crop protection, fertilization, and environmental safety through their application as pesticides, fertilizers, plant growth regulators, soil amendments, water treatment agents, food preservatives, biocontrol agents, biosensors, drug delivery systems, and environmental remediation agents [129]. They safeguard beneficial organisms and the environment through their specifically targeted action [3]. Thematic applications of nano-agrochemicals that improved synthetic pesticide efficacy are shown in Figure 4b. The effectiveness of various nano-agrochemicals in agriculture is discussed in Table 3.

### 3.1. Types of Nanopesticides (NPCs) 

NACs (such as NPCs) represent a remarkable advancement in contemporary agricultural practices. Nanostructures are used as carriers for agrochemical AIs [123]. The utilization of nanotechnology in agrochemicals effectively addresses the limitations associated with traditional agrochemicals, such as limited bioavailability, susceptibility to photolysis, and the potential for organic solvent pollution [13]. Researchers have conducted an extensive analysis of a large dataset consisting of 36,658 patents and 500 peer-reviewed journal articles, resulting in the identification of two prominent categories of NPCs. Type 1 NPCs consist of metals such as Ag, Cu, and Ti, while Type 2 NPCs involve the utilization of nanocarriers to encapsulate the AIs. These nanocarriers can be composed of various materials such as polymers, clays, and zein nanoparticles (NPs) [17]. 

#### 3.1.1. Type 1: Metal-Based Nanopesticides (m-NPCs)

The m-NPCs use metallic nanoparticles as active ingredients, providing numerous advantages compared to conventional chemical pesticides [130]. They are smaller, usually between a few and 200 nm [17,131]. The presence of multiple active sites for the release of bioactive molecules, ion-exchanging properties, high adsorption ability, efficient surface chemistry, high thermo-stability, and exceptional electronic characteristics in nanotechnology have led to the development of m-NPCs [131,132]. In general, metal formulations are synthesized using metal clusters or ions as nucleation centers which are interconnected by organic ligands. Metal-based encapsulation exhibits advantageous properties such as high surface/volume ratio, voluminous pores, adjustable pore size, efficient surface chemistry, high thermo-stability, and multiple topologies [131]. These NPCs are effective against a wide range of pests and diseases, but they also have the potential to be toxic to non-target organisms. Recent studies have demonstrated that their utilization shows superior efficacy compared to their non-nanoscale analogues. This advancement has proven to be involved in augmenting agricultural productivity, ensuring the safety of food products, and enhancing their nutritional content [17]. m-NPCs can act as pesticides by producing reactive oxygen species (ROS), releasing cations, damaging biomolecules, depleting ATP, and interacting with membranes [133]. A range of metallic and metallic oxide NPs have shown promising sustainable agriculture and antimicrobial applications in vitro against Gram-positive and -negative bacteria. Gold and silver NPs are useful for combating bacteria [15,134] and copper NPs can be used to treat fungal infections in plants [134,135] such as against pathogenic fungi *Stachybotrys chartarum* and *Candida albicans* [136]. The application of Ag NPs is their use as an antibiofilm coating. Additionally, these NPs have demonstrated antimicrobial properties against various microorganisms such as fungi and viruses, including SARS-CoV-2 [137]. Other metallic oxide NPs, like titanium dioxide (TiO_2_), zinc oxide (ZnO), and iron oxide (Fe_2_O_3_), have antimicrobial and fungicidal properties, making them useful for averting plant diseases [15]. ZnO-NPs have demonstrated enhanced efficacy against a range of microorganisms, including *Bacillus subtilis*, *Bacillus megaterium*, *Staphylococcus aureus*, *Sarcina lutea*, *Escherichia coli*, *Pseudomonas aeruginosa*, *Klebsiella pneumonia*, *Pseudomonas vulgaris*, *Candida albicans*, and *Aspergillus niger* [138]. Green synthesis of ZnO-NPs led to several morphological and histological abnormalities in *Ae. Aegypti* third instar larvae [122]. Some of the different types of m-NPCs are summarized in Table 3.

#### 3.1.2. Type 2: Nanocarrier-Based Nanopesticides (nc-NPCs)

The nc-NPCs represent a category of NPCs wherein AIs are encapsulated within nanocarriers. The AIs in this type are mainly conventional pesticides, such as atrazine, avermectin, and glyphosate. Nanocarriers can be composed of diverse materials, such as polymers, lipids, and proteins. The utilization of nanocarriers in NPCs presents numerous benefits, such as enhanced solubility, stability, and regulated release of AIs. The application of these agents may enhance the efficacy of NPCs by improving their capacity for permeability and absorption into plant tissues. Therefore, they have demonstrated enhanced action and efficacy of AIs in comparison to conventional formulations, owing to their diminutive dimensions and substantial surface area [17,131]. Recent advancements in biopolymer modification have enabled enhanced control over nanocarriers’ characteristics and their interactions with cargoes and plant tissues. Lignocellulosic-based nanocarriers offer a promising platform for the development of environmentally friendly NPCs due to their non-toxic and biodegradable nature. Tannins and β-glucan are also being studied as potential nanocarriers for AIs [139]. It has been reported that the tobacco mild green mosaic virus (TMGMV) and other plant viruses can act as nanocarriers for AIs to effectively deliver pesticides to target cells [140,141]. NPCs utilizing nanocarriers possess certain advantages; however, concerns and challenges persist regarding their application. Thorough evaluation of the risks and potential harm to people and the environment is necessary [142]. Different types of nanocarriers which have been used to encapsulate AIs (type 2 NPCs) are shown in Figure 5.

**Table 3 plants-13-00109-t003:** A summary of the composition, effectiveness, and ecological risks of NACs in agriculture.

Nanopesticide Type	Composition	Effectiveness	Ecological Risk	References
**Metal-based nanopesticides**				
Silver	Silver nanoparticles (AgNPs)	Utilized in diverse agricultural situations owing to their remarkable antimicrobial attributes.	AgNPs induce oxidative stress in plants and bioaccumulate across trophic levels, resulting in significant toxicity. They are extremely toxic to aquatic organisms.	[143]
Copper	Copper-based nanomaterials (Cu-based NMs) including Cu, Cu (I), and Cu (II)-based NMs	Promising alternative to highly active fungicides.	Excessive use of copper-based fertilizers and pesticides poses environmental risks. Cu-based nanomaterials are toxic in aquatic systems.	[144]
Zinc	Zinc oxide nanoparticles (ZnO-NPs)	Promising antibacterial, antifungal, and antiviral properties.	Harm fish and other aquatic organisms. They can cause harmful effects on genes, mutations, or cells.	[145]
Iron	Iron-based nanoparticles	Utilized as an insecticide in pest management.	Insufficient consideration of environmental and human health risks in the research.	[146]
Titanium	Titanium dioxide nanoparticles (TiO_2_ NPs)	Used in various industries due to their high photocatalytic activity.	Adverse impacts on aquatic ecosystems.	[147]
Aluminium	Nanostructured alumina	Utilized as an insecticide in pest management.	Insufficient consideration of environmental and human health risks in the research.	[146]
**Silica NPs (Silicon Dioxide NPs, SiO_2_-NPs)**				
Solid and Nonporous	SiO_2_-NPs	Insecticides, physical contact, or absorption through the insect’s cuticular layer.	Non-target organism toxicity potential.	[148]
Mesoporous	--	Damaging phytophthora infestans through intracellular peroxidation.		[149]
Spiky	--	Improved adhesion and performance of spinosad pesticide.		[150]
Nanocarriers	--	Enhanced solubility and uptake of hydrophobic agrochemicals.		[151]
With essential oils	--	Bio-efficacy on insect pests of economic and medical importance.		[123]
Silica NPs in stored grain	--	Control of stored grain pests, against two stored grain pests, *S. oryzae*, *Tribolium castaneum*, and two field pests, *Lipaphis pseudobrassicae* and *Spodoptera litura*.		[152,153]
**Nanoemulsion NPCs**				
Oil-in-Water (O/W) Nanoemulsions	Oil droplets dispersed in water. The oil:surfactant:water ratio can be 10:5:85, in volume percent.	Efficient for encapsulating and delivering lipophilic compounds with small droplet size and improved functional properties. Pesticides can also improve food quality and shelf life through biodegradable coating and packaging films.	The search results do not specifically mention the ecological risk of nanoemulsions.	[154,155,156,157,158]
Bicontinuous	Oil and water droplets are mixed together. Bicontinuous nanoemulsions form through a two-step process, starting with a bicontinuous microemulsion formation.	Providing a mechanism for the encapsulation and delivery of active ingredients.	The ecological risk of bicontinuous nanoemulsions is not specifically mentioned in the search results.	[158,159]
Water in Oil (W/O)	Consists of water droplets dispersed in oil. The ideal nanoemulsion formulation consists of 7.4% (*w*/*w*) dispersed phase (such as phenolic-rich aqueous phase from olive cake extract) and 11.2% (*w*/*w*) surfactant mixture in an oil continuous phase.	Widely used in foods, medicines, and cosmetics for the encapsulation and delivery of AIs.	The ecological risk of W/O nanoemulsions is not specifically mentioned in the search results.	[155,158,160]
Nutraceutical Nanoemulsions	Various oils, surfactants, and bioactive compounds	Enhances bioavailability of long-chain fatty acids.	Potential toxicity towards non-target organisms, needs further research.	[154]
**Polymer-Based NPCs**				
Nanocapsule	Polycaprolactone (PCL), Polyethylene glycol (PEG), Polylactic acid (PLA).	Enhanced precision and absorption, extended release, reduced chemical wastage.	Aquatic environments at risk from off-site movement.	[152,161,162]
Nanosphere	Alginic acid, gelatin, polylactic acid, chitosan, polylactide-co-glycolide, and polycaprolactone	Improved efficacy through controlled release and photo-degradation resistance, maximizing impact on target organisms. Evenly distribute AIs, enhance uptake and stability of spray solution, ensure uniform distribution.		[146,161,162]
Micelle, nanogel, electrospun nanofibers	Not specified	Not specified.	Environmental risks and future challenges are still being debated.	[161]
**Chitosan-Based NPCs**				
Chitosan–Alginate	Chitosan and alginate NPs carrying the herbicide paraquat	Efficient herbicide delivery to target plants (50–70% encapsulation efficiencies).	Lower toxicity and genotoxicity.	[163,164]
Chitosan-coated mesoporous silica	Chitosan-coated mesoporous silica NPs	Reduced disease and boosted fruit yield in watermelon seedling leaves (27% disease decrease, 70% fruit yield increase).	43.1% toxicity reduction in comparison to non-nanoscale analogues.	[17,152]
RNAi-Chitosan	Chitosan NPs used in the synthesis of RNAi-chitosan NPCs	Efficiently controls forest insect pests and microbes.	Ecological impact is low because it is biocompatible, biodegradable, and non-toxic.	[165,166]
Chitosan NPs encapsulating spinosad	Chitosan NPs encapsulating Spinosad	Not specified.	Not explicitly mentioned in the search results.	[123]
**Nanocapsules (NCs)**				
Metal-based NCs	Metal-based NPs (Ag, Cu, Ti) encapsulating AIs	Not specified.	Potential risk to human health from occupational exposure.	[17]
Polymer- and clay-based NCs	Nanocarriers (polymers, clays, zein nanoparticles) encapsulating AIs	Non-toxic to soil biota and the rhizosphere microbiome.	Potential risk to human health from occupational exposure.	[17]
Dual-functionalized pesticide NCs	NCs loaded with two AIs, validamycin and thifluzamide	Effective against *Rhizoctonia solani* at 0.0082 μg/mL.	Not specified.	[167]
Nano-emulsions based on lipids	Lipid-based nano-emulsions encapsulating essential oils of citronella and neem	Higher efficacy than classic insecticides.	Not specified.	[168]
Polymer-based NCs	NCs composed of natural polymers like chitosan, cellulose, and polylactide	Enhanced formulation, simplified application, precise pest targeting, heightened efficacy, reduced application rates.	Ecological risk is not mentioned in the search results, but concerns exist about novel products and their environmental impact.	[17,123,152,162]
Clay-based NCs	NCs composed of clay minerals like bentonite, smectite, chaolite, and montmorillonite	Enhanced efficacy, safety, and stability of agrochemicals for longer durations.		[123,152]

(a)Silica:

Silicon’s ability to improve plant tolerance to different stresses has been well established. As a result, silica nanoparticles have been proposed as potential tools for better pest management in agriculture [169] and later investigated for their potential in delivering pesticidal effects [151]. Silicon dioxide NPs (SiO_2_-NPs) exhibit a porous structure, exceptional surface activity, and notable adsorption properties, rendering them highly suitable for diverse applications, including their potential utilization in nanopesticides (1). This porosity enhances the contact between pesticides and siliceous frameworks, potentially improving their effectiveness by increasing UV-shielding capabilities [151], and lets them serve as an excellent nanocarriers for different agrochemicals [130]. They can be used in agriculture in two ways: as direct field application pesticides, killing insects, and as carriers for different herbicides and insecticides due to their ability to enhance the longevity and effectiveness of various commercial pesticides [170]. However, their efficacy as pesticides may vary depending on their origin and composition. For example, the efficacy of a compound derived from crystalline silica samples was found to be lower in suppressing potato tuber moth when compared to that derived from amorphous silica powders [171]. Studies have demonstrated that this type of NP can exert toxic effects on non-target organisms. For instance, *Galleria mellonella* larvae exposed to SiO_2_-NPs exhibited a notable reduction in both total hemocyte count and hemocyte viability [172]. Nanotubes containing aluminosilicate have been found to attach to plant surfaces and insect hair, allowing them to enter the insect body and disrupt its physiological functions. They caused 100% mortality in the cowpea weevil *C. maculatus* when applied at a rate of 2.06 g/kg. Chlorpyrifos-loaded SiO2-NPs (Ch-SNPs) were found to effectively control *R. dominica* and *T. confusum*, with mortality increasing as the concentration of Ch-SNPs increased [130]. These NPs derived from *Alstonia scholaris* exhibited increased toxicity against *R. dominica*, as evidenced by an LC_50_ value of 0.8 mg/mL and an LC_95_ value of 1.95 mg/mL. The repellent properties of NPs when combined with the plant oil *Ricinus communis* also increased against *T. castaneum* [130]. Table 3 provides a general overview of some examples of silica nanopesticides and their applications. The specific properties and applications of silica nanopesticides can vary depending on the type of nanoparticle and the target organism.

(b)Nanoemulsions:

Nanoemulsion-based pesticide formulations refer to a specific category of pesticide formulations that involve the integration of AIs within a nanoemulsion system. They are colloidal dispersions that consist of extremely small particles, typically within the size range of 20–200 nm. These emulsions are economically cheaper and commonly composed of oil-in-water (O/W type) phases [130,173]. The purpose of these formulations is to mitigate and manage the impact of pests and diseases on crops. They have been specifically developed to optimize efficacy by functioning as a carrier to transport and administer bioactive compounds to the intended pests in agricultural settings [173]. Their notable benefit is cost-effectiveness due to their high water solubility, allowing them to easily dissolve hydrophilic and lipophilic compounds. Consequently, a reduced amount of AIs and inert material is needed. The improved solubility and absorption of these formulations lead to enhanced efficacy against pathogenic organisms such as bacteria, fungi, and insects [173,174]. This potentially contributes to the reduction of environmental pollution [123]. In addition, nanoemulsions show excellent storage stability over a wide temperature range (−10 to 55 °C). They have demonstrated their effectiveness in combating various storage pests, including adults and larvae [130].

SiO_2_-NPs are employed as carriers in nanoemulsions, which is an example of utilizing mesoporous SiO_2_-NPs as delivery systems for hydrophobic substances such as drugs and pesticides. These SiO_2_-NPs improve the stability and efficacy of the nanoemulsion, specifically in delivering AI to pests in agriculture. They, with the aid of SiO_2_-NPs, have shown promise in enhancing the delivery and effectiveness of lipid-soluble substances in pesticide formulations [174,175]. Another example is β-cypermethrin, which has been successfully integrated into nanoemulsions through the utilization of different surfactants and oil phases, leading to the formation of stable pesticides exhibiting enhanced characteristics [174]. Their application in formulations has demonstrated potential for enhancing pest control efficacy, particularly with regard to insects that commonly infest stored grains [130]. Neem oil-containing nanoemulsions have proven effective against two economically significant agricultural pests: the red flour beetle (*Tribolium castaneum*) and the rice weevil (*Sitophilus oryzae*) [176]. In terms of pest control in particular, nanoemulsions present a promising alternative that can improve safety in human health and environmental aspects with the least amount of harm to the environment and non-targeted organisms [176,177]. Table 3 summarizes the types, composition, effectiveness, and ecological risks of the specified nanoemulsions. Further investigation is required to fully comprehend their potential ecological risks and their impacts on non-target organisms [154].

(c)Polymer-Based Nanopesticides (PB-NPCs):

PB-NPCs employ polymeric nanoparticles as carriers for active ingredients in pesticide formulations. NPs usually have dimensions ranging from 1 to 1000 nm [161]. The AIs are encapsulated within these polymers, which can encompass a range of agrochemicals including insecticides, herbicides, and fungicides [161,162]. Polymeric NPs possess biocompatibility, biodegradability, and the ability to undergo chemical surface modification, rendering them highly appealing for pesticide delivery [162]. NPs possess advantageous characteristics including controlled release of AIs, safeguarding against degradation, and enhanced water solubility [161,178]. Several examples exist of PB-NPCs, which serve as effective polymer nanocarriers. These nanocarriers possess desirable characteristics such as the ability to design intricate pesticide delivery systems with diverse modes of action, biocompatibility, scalability in preparation, and biodegradability [146]. Pesticide molecules are distributed randomly within a polymer matrix in nanocapsules called polymer micelles, forming a core-shell structure in polymer nanospheres. This serves as a reservoir for encapsulation [146]. Polycaprolactone (PCL), polyethylene glycol (PEG) and polylactic acid (PLA) are biodegradable polyesters utilized in the fabrication of PB-NPCs. One advantage of them is that they provide the ability to control the release of substances, as well as compatibility with biological systems. PEG is recognized for its capacity to improve the solubility and stability of AIs [161]. Chitosan, a naturally occurring polymer obtained from chitin, has been widely utilized in NPC formulations due to its abundant availability and inherent properties. It also provides benefits such as biocompatibility, biodegradability, and controlled release of active compounds [161]. The potential environmental and health impacts of PB-NPCs necessitate comprehensive evaluation and additional research to ensure their safe and effective implementation, despite their promising benefits for sustainable agriculture [161]. The composition, efficacy, and ecological risks of a few different types of PB-NPCs are summarized in Table 3.

(d)Chitosan-Based Nanopesticides (Chit-NPCs):

Chit-NPCs have garnered considerable interest in the agricultural sector owing to their distinct characteristics. Chitosan is a linear polysaccharide that is derived from chitin through the process of deacetylation. It is a naturally occurring substance. The material possesses biodegradability, biocompatibility, and non-toxicity, rendering it highly suitable for diverse applications [179]. Chit-NPCs have been found to exhibit dual functionality in agriculture, serving as both growth enhancers and potent antimicrobial agents against pathogenic fungi and bacteria. They can be developed by utilizing NPs as carriers for existing AIs [180]. The efficacy of these mechanisms can be augmented by the diminutive dimensions of the chitosan nanoformulations [180]. Silva et al. (2011) devised chitosan and alginate nanoparticles as carriers for the herbicide paraquat [164]. This intervention led to a reduction in disease incidence and a simultaneous enhancement in fruit yield [17]. Chit-NPCs have been used in the production of RNAi-Chit-NPCs (RChit-NPCs) to effectively control forest insect pests [181]. They can be used in drug delivery systems owing to their mucoadhesive characteristics, positive surface charge, and capacity to disrupt intercellular tight junctions [182]. Common techniques for the production of Chit-NPCs include ionotropic gelation, microemulsion, emulsification solvent diffusion, and emulsion-based solvent evaporation. Their particle size and surface charge are influenced by several key characteristics, including molecular weight, degree of deacetylation, pH, and chitosan concentration [166]. Table 3 summarizes the types, composition, effectiveness, and ecological risks of Chit-NPCs.

(e)Nanocapsules (NCs):

NCs are a specific class of nanopesticides that encapsulate pesticide AIs within a nanoscale shell. Similar to other NACs, these novel formulations also present many benefits compared to conventional pesticides, augmenting their effectiveness, safety, and ecological sustainability [17,123]. They can be synthesized from diverse materials such as clay minerals (bentonite, smectite, chaolite, and montmorillonite), lipids (triglycerides or waxes), inorganic porous materials, natural polymers (chitosan, cellulose, and polylactide), and synthetic polymers (polylactic acid) [17,123]. The AIs within the NCs can be incorporated into the matrix through either chemical bonding or physical adsorption, employing various techniques. This is an effective strategy to mitigate the loss of efficacy caused by evaporation, degradation, and leaching. Furthermore, it enhances the activity of substances by facilitating improved interactions with various harmful pests [123]. It shows a 31% increase in efficacy against target organisms and a 43% decrease in toxicity towards non-target organisms [183]. NCs have the potential to enhance the efficacy of pesticides through improvements in permeability, solubility, stability, and controlled release mechanisms [2].

In addition, they possess distinctive physicochemical characteristics, including adjustable dimensions, minimal cytotoxicity, and heightened efficacy of encapsulated AIs. Consequently, they can serve as a proficient vehicle for delivering AIs. This phenomenon may result in enhanced absorption by pests and heightened pesticidal efficacy [183]. Although NCs have numerous advantages, they also present certain potential challenges. Their distinctive physicochemical characteristics, which contribute to their biological impact, may also present unforeseen toxic hazards. To ensure their safe utilization, it is imperative to possess a thorough comprehension of nanoparticle toxicity [184].

However, the development and optimization of NC formulations is a complex process that necessitates specialized expertise [167]. The initial cost of NC-based pesticides may be higher compared to conventional formulations, which can be primarily attributed to the expenses involved in registering a novel AI [185]. Liposome NCs have the ability to encapsulate diverse pesticide AIs, exhibiting notable effectiveness against a range of organisms including insects, fungi, bacteria, and other pests [186]. Polymeric NCs can be synthesized using a diverse range of polymers, including both synthetic and natural polymers. They offer controlled release properties and can be modified for precise delivery [167,185]. Solid lipid nanoparticle (SLNs) NCs are composed of solid lipids, specifically triglycerides or waxes [167,185]. SLN formulations have already proven to be suitable carriers in agriculture [187]. NPs offer a sustained release mechanism for pesticides, commonly employed for foliar applications [167,185]. The efficacy of NPCs is addressed in Table 3, while an explicit discussion on ecological risks is absent in the literature. However, they are widely acknowledged for their biocompatibility and biodegradability, indicating a potentially lower ecological impact compared to conventional pesticides [17,123,162].

Semiconductor nanoparticles called “quantum dots” and carbon nanotubes are used for the targeted application of pesticides and agricultural chemicals to plants [15]. Liposomes, which are spherical vesicles made of lipids, are a precise method of delivering pesticides and agrochemicals to plants. Dendrimers in the shape of trees are effective at delivering pesticide AIs to crops with pinpoint accuracy. Nanocapsules, which are comprised of extremely small particles of polymer, can efficiently deliver agricultural chemicals to plants. Water, oil, and surfactants form nanoemulsions, which can be used to selectively deliver agricultural chemicals to plants [1,15].

### 3.2. Effectiveness of Nano-Agrochemicals

The efficacy of nano-agrochemicals in agricultural applications has been substantiated by their demonstrated ability to yield desired outcomes. They are employed to enhance the efficacy of pesticides and herbicides relative to their conventional counterparts, with the aim of enhancing disease management and pest control. They have shown to be up to ten times more toxic to their target pest than their non-nano analogues and their usage can reduce environmental contamination by 20–30% [188]. NPCs and NHCs are very effective in agriculture for nutrient and pest management due to their efficiency, high penetration ability into plant tissues or insect cuticles, and surface area due to their nano-size. These particles are environmentally friendly and effectively mitigate environmental pollution [189].

The success of these innovative agrochemicals can be attributed to a multitude of factors, such as the nanomaterial or encapsulation technique used, the pests or diseases they are designed to combat, the mechanisms underlying their functionality, and the methods by which they are applied. They demonstrate superior performance compared to conventional pesticides in various aspects [13]. Some examples of desirable characteristics in an agrochemical include the following: the ability to selectively target specific plant parts or pests for the delivery of active ingredients while minimizing off-target effects; the reduction of pesticide loss due to runoff or degradation [17,123]; the reduction of pesticide toxicity to non-target organisms; and the demonstration of synergistic effects when used in conjunction with other agrochemicals [13,190]. However, different formulations may display different functionalities, and researchers are currently investigating this. NPs’ stability, environmental interactions, and compatibility with different crop varieties must all be carefully evaluated. NACs need to be tested in the field to prove their efficacy and checked for safety before they can be used widely in agriculture [13]. The proposed antimicrobial mechanism for metal NPs is shown diagrammatically in Figure 6 [191]. It is important to remember that research is still being done to determine whether or not NACs are effective, and that different formulations may produce different results. It is critical to consider nanoparticle stability, environmental factors, and crop compatibility.

### 3.3. Ecological Risks of Nano-Agrochemicals

Ecological risks are defined as the effect and behavior of NACs on communities, populations, and ecosystems as compared to the other contaminants present in the environment. The ability of NPs to retain their properties, reactivity, and particle size when they enter the environment can make them toxic to the targeted organisms as well as non-targeted species [192,193]. As mentioned earlier, NACs have shown different effects on field crops as compared to conventional products [193]. Currently, we do not have enough knowledge to properly estimate the effects and exposure of NACs in a specific situation. It is believed that in the near future we will be able to develop a model to estimate the proper ecological risks of NACs [194]. Due to the excessive usage of NACs in the environment, their ecological risks have become a major concern in the last few years. The toxicity of NPs and NACs can be determined by their shape, size, and biodegradability. NPs can be classified on the basis of their shape and biodegradability into four categories: (i) size < 100 nm and non-biodegradable, (ii) size < 100 nm and biodegradable, (iii) size > 100 nm and biodegradable, (iv) size > 100 nm and non-biodegradable [194]. Certainly, non-biodegradable products can persist in the body and present enhanced toxicity risks. It is suggested that NPs have compound interactions with microorganisms present in the soil; even the minimum concentration of nanoparticles can disturb a microbial community [195]. Soil microbes play a very important role in the maintenance of soil ecosystems by performing different activities including nutrient recycling, growth enhancement, decomposition of soil organic matter, disease suppression, etc. Any substance that shows negative effects on microbial populations in soil may disturb the sustainability and quality of soil [130]. Similarly, a plant-associated community, *Bradyrhizobium canariense*, was revealed to be significantly sensitive to NPs [196]. Globally, the applications and safe use of nanoparticles for crop protection and yield enhancement are currently a major concern. It has been found that AgNPs showed some inhibitory effects on the activity of soil exoenzymes which enhance soil’s biochemical processes [197]. 

We can provide suggestions to resolve the challenges and concerns associated with the application of nano-agrochemicals. Conducting a comprehensive examination of the environmental fate, toxicity, and long-term consequences associated with their use is imperative prior to their widespread implementation. Regulatory frameworks are of the utmost importance, given their distinctive characteristics, and in adherence to the precautionary principle [198,199]. By adopting sustainable pest control strategies, one can effectively mitigate environmental impacts and reduce reliance on pesticides. Disseminating information regarding the advantages and disadvantages of nano-agrochemical use is crucial to fostering well-informed public discourse and ensuring responsible progress [200]. The implementation of nanocarriers to deliver pesticides specifically to pests has the potential to mitigate ecological risks and decrease environmental exposure [17]. 

## 4. Current Research Status of Nano-Agrochemicals 

NPCs and NHCs are still being developed and some of them have been on the market for many years, consisting of reformulations of registered active ingredients (AIs) with insecticidal, fungicidal, or herbicidal properties. Nanocarriers are often ‘soft’ nanoparticles (polymers, solid lipid) but there are also examples of ‘rigid’ nanomaterials such as silica nanoparticles, carbon nanotubes, or graphene oxides [201]. NPCs can serve as a complementary addition to conventional agricultural techniques, including seed coating, root irrigation, and spraying. They typically possess the advantage of being applicable through conventional spraying methods, thereby eliminating the need for specialized equipment. Given the assumption of reliable and high-quality preparation of NPCs, the implementation of these substances in practical agricultural settings is not impeded by significant technical obstacles. This research field and its development show considerable potential for future advancements. In recent years, a number of structural shifts have become apparent. The primary goals of relevant organizations encompass the optimization of operational efficiency, the mitigation of environmental impact, and the enhancement of distribution strategies [192,197]. 

The key domains of investigation and implementation encompass the following: (a) The utilization of nanocarriers, such as NPs, nanocapsules, and nanogels, facilitates the augmentation of bioavailability and the achievement of targeted delivery. Therefore, diminished levels of pesticides demonstrate heightened effectiveness through enhanced availability [202,203]. Similarly, cutting-edge methodologies have been devised in the realm of nanoencapsulation, facilitating the attainment of precise and protracted release capabilities [202,203]. Further investigation is required in order to ascertain whether NPCs present health or environmental hazards in comparison to traditional pesticides, with particular focus on their potential toxicity, degradation, and environmental dynamics [142]. (b) Nanotechnology can optimize the solubility of hydrophobic pesticides, thereby enhancing their efficacy and distribution. By leveraging the capabilities of biodegradable nanomaterials and implementing precise regulation of bioavailability, it becomes possible to achieve minimized off-target effects [202,203]. This is because they have the inherent capability to eliminate not only their targets but also beneficial insect species. Therefore, it is imperative to possess a profound understanding of the intricate mechanisms governing plant absorption and translocation phenomena [136]. (c) NPCs possess diverse mechanisms of action and a broad range of effectiveness and can also raise the issue of pest resistance [137]. There needs to be more research done to understand the lasting impacts of NPCs on delicate ecosystems and complex food webs [142]. The development of a comprehensive regulatory framework is critical in light of the upcoming commercialization of NPCs because this will maximize the benefits of this innovation while minimizing the potential drawbacks [142].

## 5. Future Prospects

However, there are several challenges and areas for further research in the field of NPCs. The following are some examples: (i) It is expected that RNA NPCs will be formalized and put into use within the next one to two years. The active ingredient is based on micro-RNA interference (MiRNA), which is double-stranded RNA or small interfering RNA, distinguishing RNA NPCs from conventional NPCs. These RNA molecules specifically target harmful organisms’ critical genes. The combination of nano delivery systems and a bacterial RNA synthesis mechanism is expected to accelerate the production of RNA-based NPCs [96,100,202]. (ii) Intelligent agriculture can benefit from accelerated research and the development of intelligent nanocarriers. Nanocarriers that are both smart and directional are useful for delivering a wide range of exogenous plant-protection factors to their intended sites of action. They are effective in a wide range of settings and contexts thanks to features like manually controlled release, temperature sensitivity, light sensitivity, and magnetically controlled release. They can be used in the future in remote control applications on smart farms [202]. (iii) The advancement of unmanned aerial vehicle (UAV) technology for plant protection has made NPCs more accessible. At the moment, the majority of UAVs used for plant protection only have small-caliber centrifugal sprinklers. Traditional pesticides are notorious for clogging and wearing out sprinklers, reducing pesticide spraying effectiveness and limiting the use and popularity of UAV technology for plant protection. Plant protection UAV operations can benefit from NPC properties [138]. (iv) The utilization of nano biosensors enables the real-time detection and monitoring of pests, leading to a more precise application of pesticides and reduced wastage. (v)The projected rise of commercialized nano-enabled formulations in the next decade to fifteen years is attributed to the increasing regulatory acceptance and continuous safety research in this field [204]. (vi) The emergence of “all-organic” nanoinsecticides is hailed as a groundbreaking strategy to mitigate adverse impacts and advance ecofriendly green agriculture. Nevertheless, our understanding of their intricate composition and practical application remains constrained, necessitating additional investigation in this domain [205].

## 6. Conclusions

The global market for pesticides is expected to increase to a value of USD 82.9 billion by 2027; the environmental and human health issues stemming from their use demand natural, green, and sustainable alternatives. Biopesticides from natural sources are more environmentally friendly than synthetic chemicals. However, their efficacy and environmental impacts await further investigations for proper optimization of the doses, species specificity, physical environmental conditions, and application. Nano-agrochemicals have become a promising substitute due to their unique properties, such as easy access, inexpensiveness, mode of action, environmental friendliness, improved efficacy with low dosage requirements, and captivating nano size. They could revolutionize agriculture, boost crop yields, and reduce environmental impact. Nanopesticides and nanoherbicides can also be tailored to agricultural needs; e.g., metal nanoparticles may kill microbes by releasing reactive oxygen species, disrupting membranes, and interfering with cellular processes. They have promising prospects, but their environmental risks and status need further investigation and thorough optimization. Understanding and managing these risks requires ongoing research and continuous monitoring. Consequently, nano-agrochemicals can improve food systems and agriculture, help achieve the SDG-17 goal of reducing hunger and poverty, and compensate for agrochemical demand globally.

## Figures and Tables

**Figure 1 plants-13-00109-f001:**
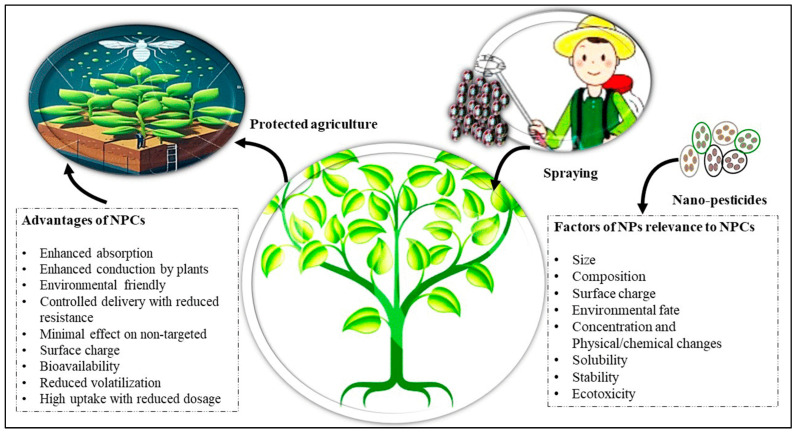
Pictorial representation of factors pertaining to NPCs and the inherent advantages they offer in protecting agricultural ecosystems.

**Figure 2 plants-13-00109-f002:**
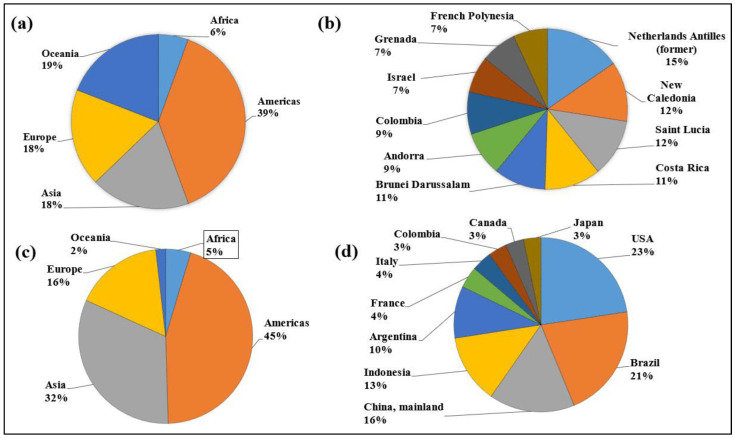
Average usage of pesticides from 2000 to 2021 [26]. (**a**) Average use per area of cropland by continent, (**b**) World’s largest consumers (top 10 countries) of pesticides per area of cropland (kg/ha), (**c**) Average percentage of use for agriculture, (**d**) World’s largest consumers (top 10 countries) for agriculture use.

**Figure 3 plants-13-00109-f003:**
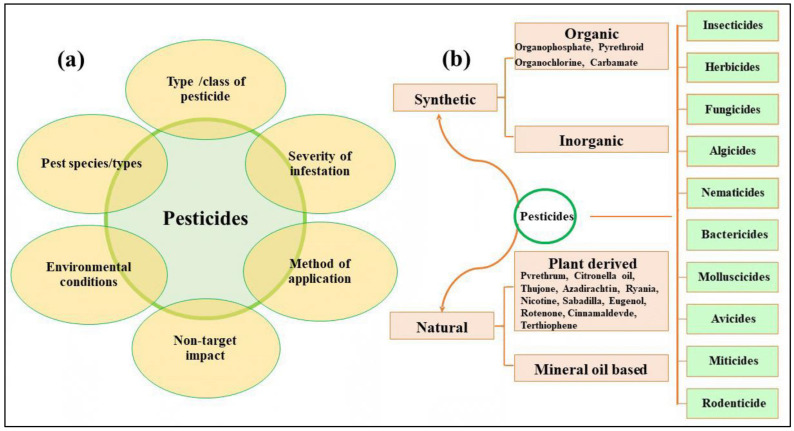
(**a**) Factors affecting the effectiveness of pesticides, (**b**) Classification of pesticides according to nature (light orange color), classification according to application (light green color).

**Figure 4 plants-13-00109-f004:**
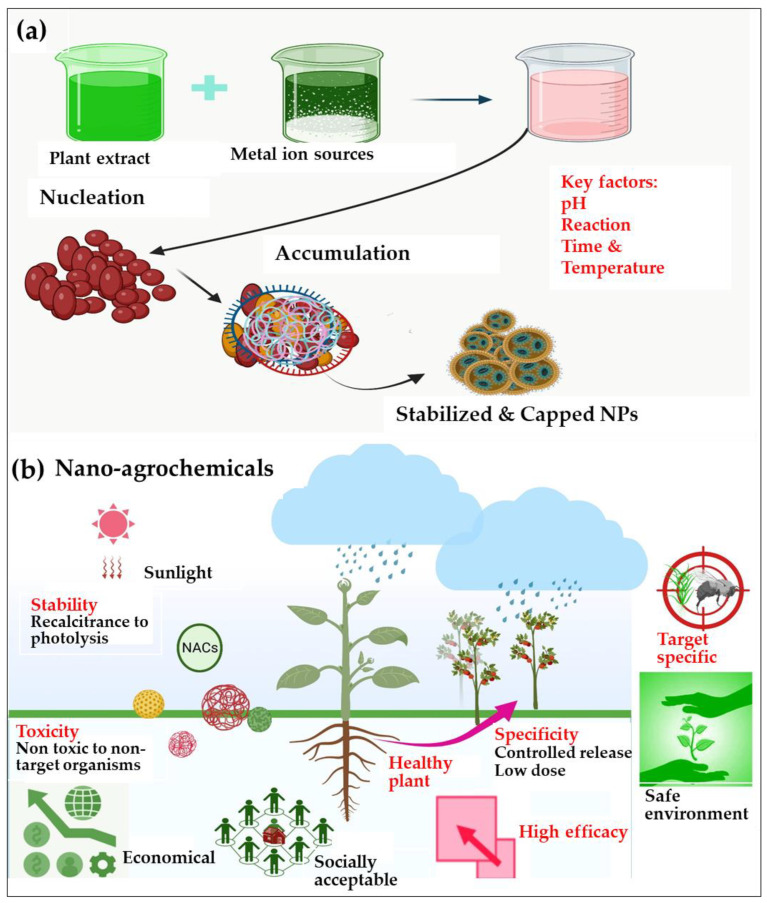
Schematic diagram for the (**a**) green synthesis of nanoparticles, (**b**) application of NACs as NPCs and improvement of the efficacy of synthetic pesticides.

**Figure 5 plants-13-00109-f005:**
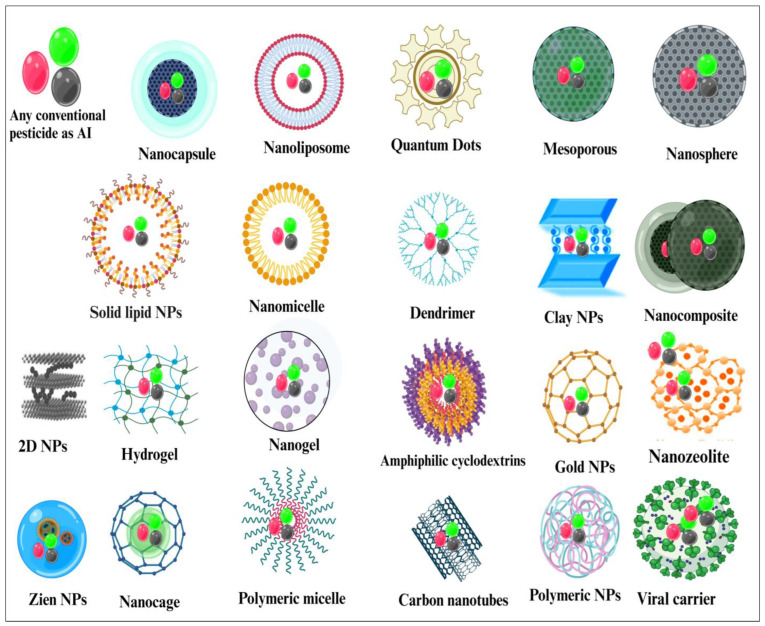
Schematic diagram of different types of nanocarriers which have been used to encapsulate AIs (type 2 NPCs).

**Figure 6 plants-13-00109-f006:**
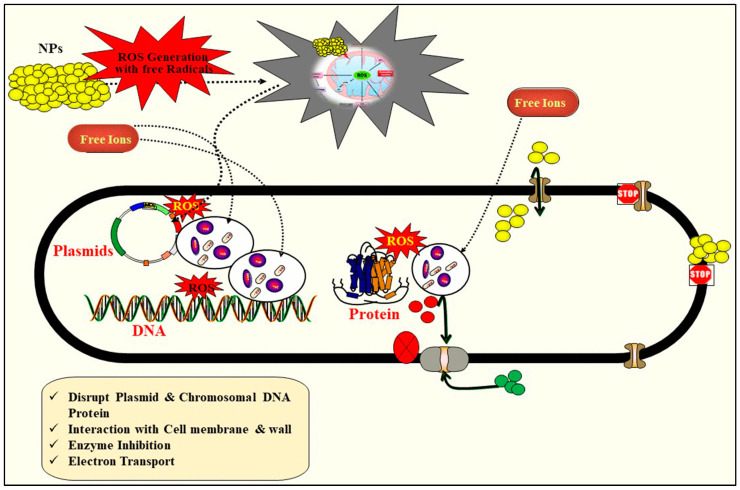
A diagrammatic representation of the proposed antimicrobial mechanism for metal NPs.

**Table 1 plants-13-00109-t001:** Effectiveness of different chemical pesticides and herbicides in agriculture.

Pesticides	Target Pest	Effectiveness	Ecological Risks/Side Effects	References
**Herbicide**				
Glyphosate	Weeds	Widely used herbicide effective against a broad spectrum of weeds	Potential impact on non-target plants and the environment	[37]
2,4-D (2,4-dichlorophenoxy acetic acid)	Turf and no-till field crops	Broadleaf herbicide used in turf and no-till field crops	Can be toxic to certain plants and should be used with caution to avoid damage to non-target vegetation	[38]
Acetochlor	Grasses and broadleaf weeds	Used to control grasses and broadleaf weeds in field corn, soybeans, and other crops	Can be harmful to aquatic organisms and should be used with care to prevent environmental contamination	[10]
Dicamba	Weeds	Effective against broadleaf weeds and woody plants	Causes damage to sensitive crops and plants, leading to regulatory restrictions and controversies	[39]
Pendimethalin	Grasses and weeds	Pre-emergent herbicide used to control annual grasses and certain broadleaf weeds in various crops	Harmful if ingested and should be handled with care to prevent accidental exposure	[40]
Trifluralin	Grasses and weeds	Effective against annual grasses and broadleaf weeds	Can contaminate groundwater and harm aquatic life	[41]
Metribuzin Oxyfluorfen, Prometryn, Pronamide. Pyraflufen-ethyl	Weeds	Effective at controlling a variety of broadleaf weeds	Can damage crops if applied incorrectly	[42,43,44,45,46]
Paraquat	Weeds	Highly effective at killing a wide range of weeds, but is non-selective, meaning it will kill any plant it comes into contact with	Can be harmful to humans and animals if ingested or absorbed through the skin, neurotoxicity	[47,48]
Isoproturon	Herbicide	Controls annual grasses and broadleaf weeds in cereals	Has been banned in EU since September 2017 due to concerns about its environmental impact and potential risks	[49]
Amitrole	Perennial grasses and broadleaf weeds (Herbicide)	Used in non-agricultural areas such as industrial lands, roadsides, railways, and ditches	Has been banned in EU after September 2017 due to concerns about its environmental impact and potential risks	[49]
Dalapon	Perennial grasses, such as quackgrass, Bermuda grass, Johnson grass, cattails, and rushes	Applied to a variety of crops including sugarcane, sugar beets, fruits, potatoes, carrots, asparagus, alfalfa, and flax, as well as in forestry	Relatively non-toxic to mammals, birds, and fish but moderately toxic to honeybees	[50,51]
Tricholoacetic acid	Grasses, sedges, and broadleaf weeds	Applied to a variety of crops including sugar beet, sugar cane, and canola	Corrosive to the skin and eyes	[50,52]
**Insecticide**				
Pyrethroids	Mosquitoes	Highly effective at killing mosquitoes	Can harm bees and other beneficial insects	[53]
Imidacloprid	Fleas	Highly effective at killing fleas and preventing them from reproducing	Can contaminate water sources and harm aquatic life	[54]
Dichlorodiphenyltrichloroethane	Insecticide	Used in agriculture and for disease vector control	Risk of breast cancer, cardiometabolic issues such as insulin resistance, impaired glucose tolerance, and high blood pressure, and increased risk of obesity	[55,56]
Permethrin, Fipronil, Carbaryl, Chlorpyrifos, Diazinon, Malathion	Ticks, cockroaches, scale insects, thrips, mealybugs, leafminers, respectively	Highly effective at killing ticks and preventing them from transmitting diseases	Can cause skin irritation and respiratory problems	[57,58,59,60,61,62]
Acetamiprid	Aphids, whiteflies, and leafhoppers	Effective against sucking insects such as aphids, whiteflies, and leafhoppers in various crops. It is considered to have low toxicity to birds, mammals, and aquatic organisms	High potential for bioaccumulation and is highly toxic to birds and moderately toxic to aquatic organisms when used excessively	[63]
Acetophos, Acephate	Whiteflies, caterpillars, beetles, and aphids	Controls a variety of pests, including caterpillars, beetles, whiteflies and aphids	Can be harmful to aquatic organisms and should be used with care to prevent environmental contamination	[64]
Aldicarb	Nematodes, mites	Effective against various insect pests in crops such as cotton, potatoes, and citrus fruits	Highly toxic to birds, fish, and bees, and its use requires strict adherence to safety guidelines	[65]
Benzene hexachloride (lindane)	Aphids, mites, and other insects	For seed treatment, in the treatment of head and body lice, in pharmaceuticals, and in the treatment of scabies	Toxic effects, including seizures, ataxia, confusion, and acute hepatorenal decompensation	[55,66]
**Fungicide**				
Thiophanate-methyl	*Botrytis cinerea*	Applied to tomato, wine grapes, beans, wheat, and aubergine. It is commonly used to treat botrytis bunch rot and gray mold caused by *Botrytis cinerea* in strawberries. Thiophanate-methyl acts as a fungicide via its primary metabolite carbendazim	Low acute toxicity, but causes liver and thyroid effects in animal studies and has been classified as a probable human carcinogen	[67]
Azoxystrobin	Fungi	Commonly used in agriculture for disease control in cereals and soybeans	Can cause skin and eye irritation and is highly toxic to certain aquatic organisms	[68]
Cyproconazole	Fungi	Controls diseases in cereals and soybeans	Adverse effects on both the environment and human health	[69]
Chlorothalonil	Fungi	Broad-spectrum fungicide used in a variety of crops, effective against many types of fungi	Non-toxic to birds but highly toxic to fish	[70]
Propiconazole	Fungi	Broad-spectrum and systemic disease control for turf and ornamentals; is also a flare root-injected systemic fungicide for control of selected diseases in trees	Possible human carcinogen, and its toxicology database indicates that the primary target organ for toxicity in animals is the liver	[71]
Dicloran	Fungi	Variety of fruits, vegetables, conifers, and ornamentals	Possible contribution to mutagenic activity	[72]
Carbendazim	Fungi	Employed to control plant diseases in cereals, fruits, and vegetables, including citrus, bananas, strawberries, pineapples, and pome fruits	Causes infertility and damages the testicles of laboratory animals	[73,74]
Copper-based fungicides	Fungi	Effective against late blight and downy mildew diseases	Excessive quantities can be harmful to plants, animals, and the environment	[75,76]
**Other important pesticide classes**				
Dichlone	Fungicide and algicide	Applied to fruits, vegetables, field crops, ornamentals, and residential and commercial outdoor areas	High exposure can cause symptoms such as headache, nausea, vomiting, dizziness, drowsiness	[77]
Dichlorophen	Fungicide, herbicide, bactericide, and algicide	Used in the treatment of small mammals	Low mammalian toxicity but moderately toxic to fish, aquatic invertebrates, and algae	[78]
Diuron	Herbicide and algicide	Annual and perennial broadleaf and grassy weeds	Cause liver enlargement, spleen and thyroid effects, red blood cell destruction, and reduction of the blood’s oxygen-carrying capacity, leading to weakness or shortness of breath	[79]
Endothal	Herbicide and algicide	Used for the control of a wide variety of terrestrial and aquatic plants	Ranges from dermal and eye irritation to respiratory failure and hemorrhaging of the gastrointestinal tract upon exposure to high concentrations for a short period of time	[80]
Fentin	Fungicide and pesticide	To control blights on potatoes, leaf spot diseases on sugar beets, and anthracnose on beans	Toxic to aquatic organisms and can persist in the environment, posing a risk to non-target species	[81]
Sodium Carbonate Peroxyhydrate	Algicide and fungicide	The active ingredient in certain algicide and fungicide products	Mild toxicity from oral and dermal exposure, but can cause dermal irritation and severe irreversible eye damage	[82]
Isothiazolines	Bactericides and algicides	Utilized in various industrial products and water treatment chemicals due to their effectiveness in controlling bacteria and algae	Can lead to allergic contact dermatitis, high acute and chronic toxicity to aquatic life, indicating potential harm to aquatic organisms	[83]
Warfarin	Mice and rats (rodenticides)	Kills rodents, such as mice and rats	Hives, rash, itching, difficulty breathing or swallowing; swelling of the face, throat, tongue, lips, or eyes are signs of an allergic reaction; risk of severe bleeding, gas, abdominal pain, bloating, changes in taste, hair loss, feeling cold, and chills	[84]
Chlorophacinone	Mice and rats (rodenticides)	Kill rodents, such as mice and rats	Can irritate and burn the skin and eyes, and can lead to chronic health effects including anemia resulting from severe or repeated bleeding	[84]

## Data Availability

Not applicable.

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
