# Peer review of "Nano-Agrochemicals as Substitutes for Pesticides: Prospects and Risks"

_plants, 2023, doi:10.3390/plants13010109_

Round 1

Reviewer 1 Report

Comments and Suggestions for Authors

From the review title I didnt expect to read this body text. A main review paper regarding nanoparticles is not included : Benelli, G. (2018). Mode of action of nanoparticles against insects. Environmental Science and Pollution Research, 25(13), 12329-12341.

For nanoparticles there are plenty of papers regarding different insect species such as mosquitoes, stored product insect species etc. There is not any evidence of all this literature as is needed. I didn't show any paper related to new nanomaterials against insect species such as graphene, silica, alumina, silver. There are plenty of papers regarding nanotechnology in agriculture

Athanassiou, C. G., Kavallieratos, N. G., Benelli, G., Losic, D., Usha Rani, P., & Desneux, N. (2018). Nanoparticles for pest control: current status and future perspectives. Journal of Pest Science, 91, 1-15.

Lai F, Wissing SA, Mu¨ller RH, Fadda AM (2006) Artemisia arborescens L. essential oil-loaded solid lipid nanoparticles for potential agricultural application: preparation and characterization. AAPS PharmSciTech 7(1):2

Goswami A, Roy I, Sengupta S, Debnath N (2010) Novel applications of solid and liquid formulations of nanoparticles against insect pests and pathogens. Thin Solid Films 519:1252–1257

Joseph T, Morrison M (2006) Nanotechnology in agriculture and food. www.nanoforum.org

Kah M, Hofmann T (2014) Nanopesticide research: current trends and future priorities. Environ Int 63:224–235

I suggest a major revision of the paper including more data regarding nanoagrochemicals (as mentioned in the title).

Comments on the Quality of English Language

From the review title I didnt expect to read this body text. A main review paper regarding nanoparticles is not included : Benelli, G. (2018). Mode of action of nanoparticles against insects. Environmental Science and Pollution Research, 25(13), 12329-12341.

For nanoparticles there are plenty of papers regarding different insect species such as mosquitoes, stored product insect species etc. There is not any evidence of all this literature as is needed. I didn't show any paper related to new nanomaterials against insect species such as graphene, silica, alumina, silver. There are plenty of papers regarding nanotechnology in agriculture

Athanassiou, C. G., Kavallieratos, N. G., Benelli, G., Losic, D., Usha Rani, P., & Desneux, N. (2018). Nanoparticles for pest control: current status and future perspectives. Journal of Pest Science, 91, 1-15.

Lai F, Wissing SA, Mu¨ller RH, Fadda AM (2006) Artemisia arborescens L. essential oil-loaded solid lipid nanoparticles for potential agricultural application: preparation and characterization. AAPS PharmSciTech 7(1):2

Goswami A, Roy I, Sengupta S, Debnath N (2010) Novel applications of solid and liquid formulations of nanoparticles against insect pests and pathogens. Thin Solid Films 519:1252–1257

Joseph T, Morrison M (2006) Nanotechnology in agriculture and food. www.nanoforum.org

Kah M, Hofmann T (2014) Nanopesticide research: current trends and future priorities. Environ Int 63:224–235

I suggest a major revision of the paper including more data regarding nanoagrochemicals (as mentioned in the title).

Author Response

The response file attached herewith

Reviewer 2 Report

Comments and Suggestions for Authors

Author Response

The response file is attached herewith

Reviewer 3 Report

Comments and Suggestions for Authors

The nano-agrochemicals as substitutes of pesticides and herbicides: Prospects and risks

Authors Ali et al. summarized current studies on nano-agrochemicals as next generation pesticides. It is an interesting review paper, but the manuscript suffers significantly large amount of structure issues. The quality of figures needs to be enhanced significantly. It will benefit to reorganize the structure of manuscript. All tables are useless to me. The topic is nano-agrochemicals; however, authors spend 12 of 19 pages to discuss current pesticide business, efficiency and biopesticides. Only 7 pages discuss nano-agrochemicals. One big flaw of this paper is that authors separate pesticides from herbicides, however, pesticides should include herbicides. The term “pesticide” is a broad category that encompasses substances used to control or eliminate pests, which can include insects (insecticides), weeds (herbicides), fungi (fungicides), rodents (rodenticides), and other unwanted organisms. Herbicides specifically target and control unwanted plants, including weeds, and therefore a subset of pesticides. I recommend authors consider revising the title, abstract, and entire manuscript including Figures 1, 2 and many sections to reflect the inclusive nature of pesticides which encompasses herbicides. This adjustment would enhance the overall clarity and precision of their work.

Following are some specific comments:

Abstract

Page 1, Change “Nano-agrichemicals” to “Nano-agrochemicals”, check the entire manuscript and keep it consistent.

Page 1, line 44, change “while decreasing environmental impact” to “while decreasing environmental contamination.” 

Introduction 

Page 1, line 50, please ensure the consistency in the numbers after the decimal point; for example, 7.35, 8.6 are not the same 

Page 1, line 93, nano agrichemicals or Nano-agrochemicals? Please keep it consistent throughout the manuscript!

Figures:                                                                          

The resolution of these figures should be enhanced significantly.

Figure 1. Nano-pesticides include nano-herbicides, this figure does not make sense!

Figure 2. This figure does not make sense as well. For example, in the figure a) pesticides should include the rest: insecticides, fungicides, herbicides and bactericides…. And where this data comes from? The data seems not right, please see the reference: Science 2013, Infographic: Pesticide Planet. The figure b), c), d) do not align with each other, indicating a potential discrepancy of data collection, and it is unclear which one represents the most accurate information to collect. 

Figure 3. this figure is very confusing too. Pesticides, insecticides, herbicides are messed up. Figure b) synthetic pesticides should be insecticides and miticides, we don’t use pyrethroids to control weeds or plant fungi.

3. alternative

3.1 Biopesticides 3.2 Bioherbicides these two sections should be combined. I feel this section 3 is not necessary for this review paper since the focus of this paper is on nano-agrochemicals.

4. Nano-agrochemicals page 12-19

This section should be expanded significantly since this is the focus of current manuscript.

Comments on the Quality of English Language

Moderate editing of English language required.

Author Response

The response file is attached herewith

Round 2

Reviewer 1 Report

Comments and Suggestions for Authors

Good to go.

Author Response

Dear Reviewer. Thank you for accepting our manuscript

Reviewer 3 Report

Comments and Suggestions for Authors

Authors modified their manuscript and enhanced the clarity significantly. I only have some minor suggestions:

Figures 1, 4, 5: keep the word fond consistently in these figures. I observed that in some figures, the height and width of the images are not locked. This means that when the document is zoomed in or out, the aspect ratio of the images may not be maintained, potentially distorting the visual representation. It would better to lock the height and width of the figures to ensure consistent and accurate display.

Author Response

The response to the reviewer's file has been attached herewith
